# Brief Communication: Recent estimates of glacier mass loss for western North America from laser altimetry

Brian Menounos[1,2,3*], Alex Gardner[4], Caitlyn Forentine[5], Andrew Fountain[6]

[1]University of Northern British Columbia, Geography Earth and Environmental Sciences, Prince George BC, V2N 4Z9, Canada
[2]Hakai Institute, Campbell River, BC, Canada
[3]Geological Survey of Canada - Pacific, Sidney, BC, Canada
[4]Jet Propulsion Laboratory, California Institute of Technology, Pasadena, CA 91109, USA
[5]United States Geological Survey Northern Rocky Mountain Science Center, Bozeman, MT, USA
[6]Portland State University, Department of Geology, Portland, OR, 97201, USA

*Corresponding author: menounos@unbc.ca

*Correspondence to*: Brian Menounos (menounos@unbc.ca)

**Abstract.** Glaciers in Western North American outside of Alaska are often overlooked in global studies, because their potential to contribute to changes in sea level is small. Nonetheless, these glaciers represent important sources of freshwater, especially during times of drought. Differencing recent ICESat-2 data from a digital elevation model derived from a combination of synthetic aperture radar data (TerraSAR-X/TanDEM-X), we find that over the period 2013-2021, glaciers in western North America lost mass at a rate of -12.3 ± 3.5 Gt yr$^{-1}$. This rate is comparable to the rate of mass loss **(**-11.7 ± 1.0 Gt yr$^{-1}$**)** for the period 2018-2022 calculated through trend analysis using ICESat-2 and Global Ecosystems Dynamics Investigation (GEDI) data.

## 1 Introduction

Western North American glaciers outside of Alaska cover 14,384 km$^2$ of mountainous terrain (Pfeffer et al. 2014). Although the global sea level equivalent of these glaciers is only $2.6 \pm 0.7$ mm (Farinotti et al., 2019), these glaciers provide important thermal buffering capacity during late summer or during times of drought (Moore et al., 2009). Early attempts to define regional estimates of glacier mass change suffered from sparse, in-situ glaciological observations, non-uniform distribution of geodetic measurements, and uncertainties in gravimetric assessments due to changes in seasonal water storage (Jacob et al., 2012; Gardner et al., 2013; Zemp et al., 2019). Two recent studies combined publicly-available geodetic datasets and statistical methods to yield mass change estimates with much less spatial bias and lower overall uncertainties (Menounos et al., 2019; Hugonnet et al., 2021). Both of these studies rely on DEMs generated from NASA's Advanced Spaceborne Thermal Emission and Reflection Radiometer (ASTER) sensor aboard the Terra satellite. Unfortunately, Terra's orbit is degrading and will reach its end of life within the next 3-4 years (https://terra.nasa.gov/). Additional datasets are thus required to quantify glacier mass loss in mountain environments where glacier loss is accelerating (Hugonnet et al., 2021), but the glaciers of western North America have so far been excluded from global altimetry assessments (Jakob and Gourmelen, 2023). Eight of the 19 regions of the globally complete Randolph Glacier Inventory (RGI) are sparsely glacierized, including Western North America. Models and current ice volume estimates suggest that these regions will each contribute ≤ 2 mm to sea level by 2100 under a +2° C global mean temperature warming scenario (Rounce et al. 2023). Several of these regions were not assessed by Jakob and Gourmelen (2023) due to the small size of the glaciers within these

regions and complex topography that makes CryoSat-2 processing challenging due in part to the larger beam diameter of
CryoSat-2 (~ 380 m) compared to IceSat-2 (~12 m). Here we provide new estimates of recent glacier mass loss based on
laser altimetry data for the western United States and Canada which is Region 02 of the Randolph Glacier Inventory (Pfeffer
et al., 2014).

## 2 Data and methods

### 2.1 Altimetric data (ICESat-2 and GEDI)

Altimetric data include observations made by NASA's Advanced Topographic Laser Altimeter System (ATLAS), which is a
532 nm photon-counting laser system aboard the ICESat-2 satellite that operates in latitudes between 88° N/S (Markus et al.,
2017). We use version 5 of the ATL06 (land-ice surface heights) dataset that includes laser shots from 13 October 2018 to 12
October, 2022. We also used Global Ecosystem Dynamics Investigation (GEDI) laser data (Liu et al., 2021) acquired
between 1 January, 2018 and 1 January, 2022 (GEDI02_A release 2). GEDI is a 1064 nm, full-waveform laser that, because
of its operation aboard the International Space Station, operates in latitudes between 51.6° N/S.

### 2.2 Digital elevation model

The mass change estimate for approximately the last decade (2013 to 2020), herein referred to as the decadal estimate, uses
the global, 30 m Copernicus DEM elevation data derived from the TanDEM-X Synthetic Aperture Radar (SAR) mission
(Rizzoli et al., 2017) and made publicly available as the Glo30 product, herein referred to as COP-30
(https://spacedata.copernicus.eu/collections/copernicus-digital-elevation-model). Acquisition of the data used in COP-30
DEM occurred between 2010 and early 2015 and coverage represented about five individual SAR tiles in our study region.
Because no gridded acquisition date exists for COP-30, we use an acquisition date of 2013, which coincides with the
midpoint for the majority of DEM acquisitions (Rizzoli et al., 2017). As described below, we use the ambiguity of DEM
acquisition dates as one source of uncertainty in our mass change estimate. Another source of uncertainty is penetration of
the TanDEM-X radar signal into high elevation firn and snow surfaces (Abdullahi et al., 2019). As described in the
discussion section of our paper, we consider the magnitude of this bias to be small.

For each subregion, we reprojected the COP-30 into the respective UTM zone of a given subregion. The COP-30 vertical
datum is EGM96 which we converted to match the vertical datum of ICESat-2 (WGS84). We clipped ICESat-2 data for a
given acquisition date to a region of interest and extracted the closest grid point of the COP-30 data for a given laser shot.
Retained data include elevation of both COP-30 and ICESat-2, derived elevation change [m] and rates of elevation change
[m yr$^{-1}$]. We also include other original attributes present with the ICESat-2 data (e.g. track number, effective laser shot
radius, slope) to maintain metadata continuity. Excluded elevation change values exceeded elevation change rates of -20 or
20 m yr$^{-1}$ since we assumed that these signals exceed the range of what is physically attributable to glacier processes. To our
knowledge, we know of no glaciers in WNA that experience surging or advance over the past two decades (Bevington and
Menounos, 2021; Fountain et al., 2023).

For the decadal estimate of mass change, we buffered each glacier polygon (RGI ver. 6.0) within the study region by 1 km
and then masked from the original glacier polygon, to capture areas adjacent to glaciers that we considered to be areas of
stable terrain. This stable terrain might include vegetated terrain, landslides or standing water, however. Due to the buffer,
we expect results to be robust to glacier polygon updates. Note that the recently released RGI-7.0 has no changes from RGI-
6.0 in our study area. Inspection of elevation change over stable terrain for all ICESat-2 laser shots (2.24 x 10$^6$) reveals a
positive bias for almost every subregion, typically on the order of 0.1-0.5 m yr$^{-1}$ (ICESat-2 minus COP-30); this bias,
however, did not substantially vary with elevation for a given region. Visual inspection of elevation change maps and review
of acquisition dates of ICESat-2 data suggests this positive bias arises by laser shots over snow-covered terrain (c.f. Enderlin

et al., 2022). We therefore limit our analysis to the ablation season when the positive bias associated with snow-covered
terrain is minimized. Confirmation of the source of this bias is revealed when the analysis of rates of elevation change is
limited to ICESat-2 laser shots acquired between 1 August and 1 October. For these late summer laser shots, we respectively
observe a mean bias and uncertainty (± 1 sigma) over stable terrain of 0.038 and 1.53 m yr$^{-1}$.

**2.3 Recent rate of elevation change from ICESat-2 and GEDI**

For the period 2018-2022, herein referred to as the recent period, we first create altimetry anomalies by differencing ICESat-
2 and GEDI laser shots to the COP-30 DEM. A least squares regression that includes an offset, trend and seasonal sinusoidal
terms is fit to anomalies within a 250 m radius search window. The y-intercept of the regression is set to the year 2020. We
exclude any ICESat-2 or GEDI laser shots if they deviate more than 250 m from the COP-30 DEM, or if they deviate by
more than 150 m from the median anomaly within the 250 m search radius. The search radius and median anomaly threshold
were selected to omit elevation change signals that were not physically realistic. Regression fits were excluded from further
analysis if: (i) there were fewer than five data point for given search window; (ii) the temporal span of observations is less
than three years; (iii) the root mean squared error (RMSE) of the fit residuals exceed 5.0 m yr$^{-1}$ and (iv); the seasonal
amplitude of the least squares fit exceeds 10 m yr$^{-1}$. We use the trend obtained from the regression to the 250 m radius to
represent elevation change.   This filtering yielded an unbiased sample across elevation bins of ice in study area (i.e. the area
distributions of sampled vs. observed ice were similar).
**2.4  Mass change uncertainty**
Uncertainty in mass change originates from errors in rates of elevation change and volume-to-mass conversion factor. We
use 850 kg m$^{-3}$ and its associated uncertainty term (±60 kg m$^{-3}$) for mass conversion (Huss, 2013). We generate bootstrapped
errors in total volume change using a Monte Carlo method. We first temporally randomize the laser altimetric data, randomly
choose the acquisition date of the COP-30 DEM (2012, 2013, 2014) and sample 5% of the data with replacement 1,000
104 times. Total volume change over glacierized terrain is calculated for each synthetic dataset by multiplying the rate of
105 elevation change by the area of glaciers within a given elevation bin (100 m bins). We then take 5% and 95% modelled
volume change as our uncertainty.
Uncertainty in mass change is then calculated from:

$$\sqrt{(dV_\sigma \cdot \rho)^2 + (\rho_\sigma \cdot dV)^2}$$

(1)

Where $dV_\sigma$ is the uncertainty of volume change generated from the Monte Carlo method, $\rho$ is material density (850 kg m$^{-3}$),
$\rho_\sigma$ is uncertainty of density (60 kg m$^{-3}$) and $dV$ is the change in volume.

**3.0 Results**

To minimize the impact of the seasonal snow signal, we limit the presentation of our analysis to mass change using ICESat-2
and COP-30 elevation changes to ICESat-2 data acquired during the latter half of the ablation season (1 August - 1 October).
Glaciers throughout the western United States and Canada thinned both during the decadal and recent period with prominent
thinning within the Southern Coast Mountains, a region that contains nearly one half of the total ice cover of the study region
(Fig. 2). For the period 2013-2021 (median date of ICESat-2 data is 26 August, 2020), we estimate a rate of mass change of
120 -12.3 ± 3.5 Gt yr$^{-1}$ (Fig. 1). This measurement agrees within the rate of mass change [−12.3 ± 4.6 Gt yr$^{-1}$] reported for the
121 period 2009–2018 (Menounos et al., 2019) and the estimate [−12.3 ± 3.0 Gt yr$^{-1}$] for the period 2015-2019 based primarily
on ASTER data (Hugonnet et al., 2021).  Comparable estimates of mass loss exist for western North America for the period
1961-2016 [-12 ± 6 Gt yr$^{-1}$] and for the period 2002-2009 [-14 ± 3 Gt yr$^{-1}$] respectively from Zemp et al., (2019) and Gardner
et al., (2013).  Using only ICESat-2 and GEDI laser shots and rates of elevation change determined through least squares
fitting (i.e. the recent period), glaciers lost -11.7 ± 1.0 Gt yr$^{-1}$ of mass for the period 2018-2022 (Fig. 2). Mass change rates

126  per subregions (Fig. 1) are summarized in the supplementary material (SM Table 1). The effect of a small sample size is
evident in the larger uncertainty of elevation change at highest and lowest elevations, but the contribution of this error to
total mass change is small since little total glacierized area exists at these elevations.

## 4.0 Discussion and Conclusion

Our geodetic balance obtained from laser altimetry using least squares fitting provides the most recent mass change update
for western North America, a region excluded in a recent global assessment of glacier mass loss using laser altimetry from
CryoSat-2 data (Jakob and Gourmelen, 2023). While our trend analysis provides a robust estimate of recent glacier mass
change, insufficient sampling precludes our assessment of mass loss for regions where laser altimetry data are sparse. This
sparseness is especially pronounced in regions north of GEDI data coverage (51.6° N), e.g. Nahanni, and regions
characterised by very small glaciers, e.g. Sierra Nevada (Fig. 2). Our decadal estimates of glacier mass loss provide insight
into sub-regional patterns of glacier mass loss, but insight is offset by the additional uncertainty of radar penetration at
highest elevation and the ambiguity of the acquisition data for the COP-30 DEM. Others report penetration of the TanDEM-
X radar signal into high elevation firn and snow surfaces (Abdullahi et al., 2019). The potential of this penetration bias to
greatly affect our results is limited since it is spatially limited to highest elevation zones containing dry snow and firn (Millan
et al., 2015); these zones typically represent < 1-2% of the total glacierized area within a given region of this study.

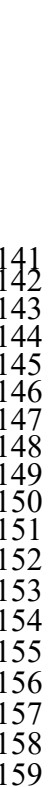

**Figure 1: Elevation change [m yr⁻¹] for western North American glaciers. Data are aggregated to points with 50 km spacing. Left panel (a): Elevation change [m yr⁻¹] determined from ICESat-2 and COP-30 data (2020 - 2013); Right Panel (b): Elevation change [m yr⁻¹] from trend analysis over period 2022-2018 from ICESat-2 and GEDI laser altimetric data. Numbers refer to glacierized regions of Western North America (RGI region 02). The regions include: (1) Central Coast (1,692 km²); (2) Southern Coast (7,181 km²); (3) Vancouver Island (15 km²); (4) Northern Interior (572 km²); (5) Southern Interior (1,959 km²); (6) Nahanni (657 km²); (7) Northern Rocky Mountains (415 km²); (8) Central Rocky Mountains (422 km²); (9) Southern Rocky Mountains (1,350 km²); (10) Olympics (30 km²); (11) North Cascades (250 km²); (12) South Cascades (153 km²); (13) Sierra Nevada (11 km²); (14) Glacier National Park (11 km²) and; (15) Wind River (60 km²).**

The regional pattern of elevation change obtained for the recent period shows areas of neutral or slight elevation gain (e.g. regions 1 and 5) that are not apparent in the map of decadal elevation change (Fig. 1). The most parsimonious explanation for these differences is the influence of spatially variable snow accumulation in these regions, though we cannot rule out the possibility of changing balance between ice dynamics and mass balance to explain the observed elevation changes. In addition, the decadal pattern largely accords with the notable zonal difference in elevation change observed by Menounos et al., (2019). A key finding of Hugonnet et al., (2021) was the notable accelerated mass loss in western North America during the period 2015-2019 relative to the start of the 21st century. Our decadal results are consistent in both magnitude and uncertainty to previous estimates using instruments (i.e. ASTER) that will soon be unavailable, and so our approach assures

mass change estimates can be obtained using much sparser observations from laser altimetry. Our recent and decadal estimates of glacier mass loss using independent datasets confirms the magnitude of recent mass change for a comparably recent period (2018 to 2022), corroborating the finding of accelerated mass loss from this previous study.

Glaciers in western North America provide cold meltwater that buffers hot and dry conditions (Anderson and Radić, 2020; Moore et al., 2009), sustains alpine stream ecosystems (e.g. Muhlfeld et al., 2020), and supports downstream communities via agricultural irrigation and hydroelectric power generation (e.g. Frans et al., 2018). Thus, our study provides relevant, detailed information to land managers who are responsible for understanding and responding to the local consequences of rapid glacier change. Sparsely glacierized regions in Western North America and Europe contribute minimally to sea level change (Rounce et al., 2023) but coincide with river basins where mountain water supply and downstream demand are highest (Immerzeel et al., 2019). This justifies the need to surmount technical and data limitations that impede quantifying glacier mass change in sparsely glacierized regions. The projected, continued loss of glacier ice (Rounce et al. 2023) furthermore suggests that this technical challenge will only become more widespread.

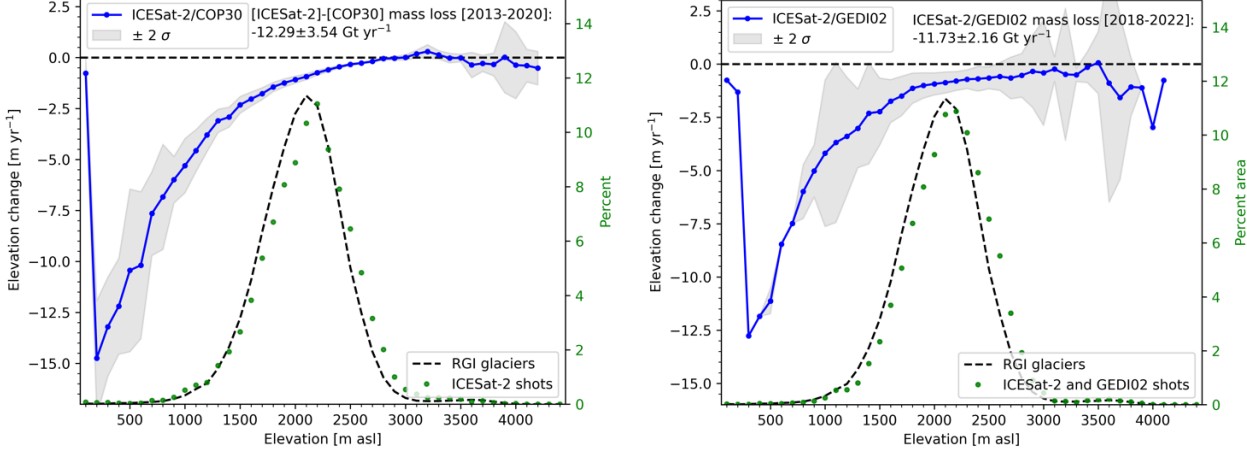

**Figure 2: In both panels, light grey shading denotes uncertainty (5-95%) of elevation change. Black dashed line and green dots, respectively, indicate percent area of RGI ice and percentage of ICESat-2 laser shots within a given elevation bin. Left Panel: Rates of elevation change [m yr⁻¹] versus elevation for the period 2013-2020. Only laser shots from 1 August-1 October (n=347,630) used in analysis. Right Panel: Rates of elevation change [m yr⁻¹] versus elevation for the period 2018-2022 from ICESat-2 and GEDI laser shots from least-squares trend analysis (n=66,201).**

**Code and data availability**

Available upon request from the authors.

**Declaration of competing interest**

The authors declare that they have no competing interests that influenced the research presented in this publication.

192

**Author contribution**

BM proposed the study. BM and AG analyzed the data and wrote the original draft. All authors provided feedback on the initial draft of the manuscript and contributed to the final writing and editing of the paper.

**Acknowledgements**
The authors acknowledge constructive input from Rainey Aberle, Albin Wells, Erik Mannerfelt and an anonymous referee which improved the quality and clarity of this manuscript. Menounos acknowledges support from the National Research and Engineering Council of Canada, the Tula Foundation. Florentine acknowledges support from the U.S. Geological Survey Ecosystem Mission Area Climate Research and Development Program. Any use of trade, firm, or product names is for descriptive purposes only and does not imply endorsement by the U.S. Government.

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
