# Peer review of "Brief Communication: Recent estimates of glacier mass loss for"

_EGUsphere, 2023_

## Author Comment (AC1)

**Our responses to the referees are listed below in bold, blue text.**

Referee 1 - Erik Mannerfelt

The authors present a new regional geodetic glacier mass balance estimate for western North America using well established methods to a study area and rationale that ensures its novelty. They argue the use of the analysis well; similar studies have already been performed but their source of data (ASTER) will not be available for many more years. Therefore, we need a new source for monitoring regional mass balance, and the authors present the solution well.

Generally, the text is concise and well-written. My comments are only textual edits, and I therefore recommend this paper for acceptance after correcting them. Some comments on the figures could either be typos, misunderstandings by me, or actual issues with the plots. Therefore, I chose "minor revisions" instead of "technical corrections" as I presume that gives some more time to fix these (potential) issues.

**We wish to thank Dr. Mannerfelt for his constructive feedback which, as described below, should greatly improve the presentation of our paper.**

**General comments:**

The tense is slightly inconsistent in the text. At L41, "we use"; first person present. At L55, "we reprojected"; first person past. L63: "was buffered"; third person past. Please look over the text and pick one for consistency.

**This is a fair point; we apologize for being inconsistent. We prefer active voice and have corrected any passive voice in the manuscript.**

The two figures are generally very informative! They can be improved to make them even better in terms of labeling, and the second figure has an unclear distinction between two black lines. I elaborate on this in my specific comments below.

**Thank you for these suggestions. We respond under those specific comments how we improved these figures.**

Is there a good reason for the data will not be made available online? The per-RGI values and rates would certainly be useful in other studies! I wholeheartedly recommend making the data available through e.g. Zenodo, and preferably the code too. I will however leave it up to the editor whether my recommendation should be enforced or not.

**We are comfortable making the elevation change data available for public access, but our general feeling is that these data won't be actively sought after since updated studies will want to ingest all available data from both laser altimetry campaigns and do their own filtering. We are happy to share our code used for post processing, but prefer individuals to contact us since some of our code is being modified for papers that currently are in review.**

**Line-specific comments:**

**L33:** "Recent studies" imply multiple studies, but only one is referred to. If there's only one, I suggest rephrasing the sentence. For example, "but the glaciers of western North America have so far been excluded from global laser altimetry assessments (Jakob and Gourmelen, 2023)."

**A good catch. Sentence is now revised to include this useful suggestion.**

**L33-34**: Why was western North America excluded by the cited paper? It is previously mentioned in the text and also mentioned in Jakob and Gourmelen (2023), but the sentence is missing a word or two, as it right now looks like an arbitrary decision by the authors of the cited paper. Honestly, their own rationale is a bit arbitrary ("[...] particularly challenging to capture with radar altimetry, including [...] USA [...]"; challenging how?). If another sentence is added to this manuscript, I think it would read better. Perhaps you can speculate that they excluded it from a cost-benefit perspective (it's "challenging", but again, why?, and it does not contribute that much to global sea level rise), as the US would not improve their results much. Or there is a good reason why they did it, but you think you can do better?

**Many of the smaller glaciated regions of the Randolf Glacier Inventory (RGI) were not assessed by Jakob and Gourmelen (2023) due to the small size of the glaciers within these regions and complex topography that makes CryoSat-2 processing challenging (Livia Jakob, personal communication). The larger footprint of CryoSat-2 (~ 380 m) compared to a nominal beam diameter of about 20 m for IceSat-2 is one of these challenges for small glaciers. Additional constraints include CryoSat-2's SARin acquisition mode for Swath processing which is not covered for some glaciers in the study area (Livia Jakob, personal communication).**

**We added, 'Many of the smaller glaciated regions of the Randolf Glacier Inventory (RGI) were not assessed by Jakob and Gourmelen (2023) due to the small size of the glaciers within these regions and complex topography that makes CryoSat-2 processing challenging due in part to the larger beam diameter of CryoSat-2 (~ 380 m) compared to IceSat-2 (~20 m).'**

**L40:** I first misread the 88 degree N/S statement to be the inclination of the orbit. If you added "in latitudes" before "between" I think it would read better.

**Added**

**L44:** Same comment as L40: I recommend specifying "latitudes between 51.6° N/S".

**Added**

**L55:** Which UTM zone and which coordinate system? There are many other coordinate systems with UTM zones. You can add a parenthesis for it "(WGS84 UTM Zone XX N)".

**Our study covers four UTM zones and hence the use of the term 'local' was to imply for given region. However, if it's not clear to a referee, it won't be clear to the reader. We revised the statement to now read,**

**'..into the respective UTM zone of a given subregion'**

**L58:** Replace the comma of "[m], rates of [...]" with an "and".

**Completed**

**L60:** Surges may occur that exceed +20 m/a at their termini! Please add a sentence explaining that you assume this to improve the total uncertainty using this filter, in spite of maybe accidentally filtering out extreme true values on surges at individual glaciers. Or if you qualitatively checked extreme areas and found that the filter was okay, please add this instead.

**This is a fair criticism since some glaciers in Alaska and northernmost British Columbia do surge. To our knowledge, however, we are unaware of any glaciers that have surged in RGI region 02 (Western Canada/US). We now state this and refer to previous work that backs up this statement.**

**L63:** This is the first time the abbreviation RGI is used. Please explain it here or add the abbreviation to L36 where Randolph Glacier Inventory is spelled out.

**Good catch. We now spell this out on L36.**

**L64-65:** Please elaborate on how you can certainly consider the surrounding terrain stable. For example, if the bedrock is generally stable and most (all?) glaciers are surrounded by bare rock. What about vegetation and lakes? Is ICESat-2 and GEDI all robust to that? If you qualitatively saw that lakes and vegetation don't seem to be an issue, that would be okay for me to state too.

**Another fair criticism. The reviewers' comment would only apply to the ICESat-2 to COP analysis since the trend analysis using ICESat-2 and GEDI laser shots was completed over ice and not stable terrain.**

**Given the area of which these buffers were applied, some vegetation, lakes and unstable terrain due to landslides might exist in these so-called 'stable areas'. However, given the low bias and spread of elevation change over these regions, we feel that our estimates provide an unbiased estimate of elevation change. If these 'stable' areas contained effects due to vegetation, land sliding or changes in lake level then they would contribute to a higher uncertainty in elevation change or, in our study, a larger standard deviation. We did observe that seasonal snow introduces a bias which we reduced by only considering summer laser shots.**

**To be fair, we now include the statement, 'This stable terrain might include vegetated terrain, landslides or standing water, however.'**

**L65**: Are the RGI or GLIMS ids the same as well? If not, it would require some work to use your data in the future.

**We checked both RGI versions and although the polygons are the same the RGI IDs differ. However, since our csv files for elevation change contain coordinates (latitude, longitude) and not RGI or GLIMS IDs for a given laser shot, this should not a problem for future use (i.e. someone could easily apply a spatial join to our data if required).**

**L70**: Add "c.f." to the Enderlin reference if you actually inspected this yourself too.

**We did and now added 'c.f.'**

**L80**: Again, I would argue that 150 m from a median anomaly is realistic in extreme cases on surges. Please argue for why this is (hopefully) not relevant. If you feel like this has been argued enough for in my L60 comment, I welcome you to ignore this one. Basically, if you at some point establish that surges will be a source of uncertainty (or are they rare enough?) and you exclude their consideration, none of this would be a problem to me.

**See previous comment on this.**

**L83**: Please rephrase "This did not disrupt the representation of glacier hypsometry" as the reader does not know how you know this. For example, starting with "We did not find a considerable disruption […] due to these criteria"

**We see the confusion of this sentence. Sentence now modified to read,**

**'This filtering yielded an unbiased sample across elevation bins of ice in study area (i.e. the area distributions of sampled vs. observed ice were similar). '**

**L88-94**: I've never seen this approach before to include uncertainties in time. I like it! It does sound computationally expensive for large regions, however. Do you have a computation time statistic to support or discourage using the same method on equally large or larger regions?

**In terms of our study, this sampling did not take a long time (10 minutes on a modern laptop with multiple (8) physical cores). Even for a large region, a Monte Carlo approach should not take an excessively long time to compute. In reviewing our analysis against what we reported in the initial submission, however, we noted a typographical error in the number of simulations performed (we stated 10,000 but the number of simulations was 1,000). Since the Monte Carlo sampling code also generates the figures, we note stability in the reported mass change estimates (to the second decimal place) based on multiple times that we executed this to draft/revise the figure.**

**L96**: I would personally stay consistent with the use of "uncertainty" instead of "error", as we have no idea what the real value is, and the word error can thus be misleading.

**A logical point that we agree with. Corrected.**

**L103**: "we limit our analysis" reads more like the methods section. If you don't want to move the sentence, I suggest changing it to "we limit the presentation of our analysis" or something like it to be more in line with a results section.

**We agree. Changed.**

**L112-113**: A word or two is missing here: "[...] recent period), glaciers lost". For example, if the sentence started with "Figure 2 shows that when using only [...]", the problem would be fixed. Alternatively, consider splitting the sentence in two.

**This was a good catch. We believe the sentence was garbled converting the file from Google Docs to a Word template for the Cryosphere. The sentence now reads,**

**'Using only ICESat-2 and GEDI laser shots and rates of elevation change determined through least squares fitting (i.e. the recent period), glaciers lost -11.7 ± 1.0 Gt yr$^{-1}$ of mass for the period 2018-2022 (Fig. 2).'**

**L114**: "elsewhere" sounds vague. Consider changing it to "the supplementary material".

**Completed.**

**L114**: A word is missing or the tense is wrong. "the effect of a small sample size" or "the effect of small sample sizes" would read better.

**Completed.**

**Discussion and conclusion**: Please add a sentence (perhaps in the very end), that your results are consistent in both magnitude and uncertainty to previous estimates using instruments (i.e. ASTER) that will soon be unavailable. Your approach therefore "secures" our need for continued up-to-date information in the future.

**A useful suggestion. These points are now added.**

**Figure 1 caption**: "Data *are* aggregated […]" will make the second sentence read better.

**Added.**

**Figure 1 caption**: To be consistent with the text, with Figure 2, and with most other publications, I would suggest switching the years to "2013-2022" instead of "2022-2013" (and with all similar occurrences).

**We decided to remove this label and, instead, add 'a' and 'b' to the captions since this label repeated text within the caption and deemed redundant.**

**Figure 1**: The forward slash (e.g. "ICESat-2/COP-30") reads a bit like there's division going on. I presume that this is because of the hyphens in the product names, but I would suggest trying out an en-dash or the word "to" for clarity.

**See above.**

**Figure 2 caption**: Why is this 2013-2021 while Fig. 1 is 2023-2022? If there is a good explanation and this is not a typo, please elaborate in the caption to make it easier for the reader.

**Sorry about that. Yes, that was a typographic error.**

**Figure 2**: Please change the legend labels to be more readable and consistent with the text and Figure 1. "Copdem30" → "COP-30", "rgi_glaciers" → "RGI glaciers".

**Completed.**

**Figure 2**: It looks to me like the "rgi_glaciers" and "IS2 shots" lines are offset horizontally by, say, half a bin. Is this an error? If not, please disregard this comment.

**We don't believe this is offset, but a minor difference in area-altitude distributions.**

**Figure 2**: The elevation change and "rgi_glaciers" lines are very similar. I would recommend changing one of these lines' color to make the legend and the plot easier to understand.

**We changed the elevation change line to blue (solid) and the RGI glacier line to a dashed one.**

---

## Author Comment (AC2)

**Our responses to the referees are listed below in bold, blue text.**

Referee 2 – Anonymous

Review of Menounos et al.

I have reviewed the submitted manuscript and believe that this is a nice complementary study that expands on the current understanding of mass change in Western Canada. The text would largely benefit from some further clarification and correction of some inconsistencies and after some minor revisions, suggested below, I would endorse this for publication.

**We thank Referee 2 for taking time to review our brief communication and for providing helpful feedback that we feel strengthens our paper.**

L15: Alaska --> would it be better to change this to Yukon/Alaska (or Alaska/Yukon)? Many global assessments of glacier mass balance term the region Alaska but the actual areas tend to include regions that fall within the Yukon as well (and perhaps portions of Northern British Columbia).

**We see the issue that the referee raises here, but given the precedent on the usage of the term 'Alaskan Glaciers' we prefer to use Alaska. To our knowledge, Arendt et al., 2002 (Science, 297(5580):382-6. doi: 10.1126/science.1072497) was one of the first to use this term to include Alaskan glaciers and those on the eastern Alaskan border. The glaciological community has largely adopted the naming convention of the Randolf Glacier Inventory (RGI) in which region 01 (Alaska) represents Alaskan glaciers and those that straddle the Alaska-BC, Alaska-Yukon borders. Since we use glacier extents from the RGI, we wish to maintain terminology that is consistent with this global mapping exercise.**

L23: Same comment as above.

**Please refer to our previous comment.**

L31-32: Regarding Terra's orbit, is there a study or technical document that you can cite here for this statement?

**We now provide the Terra website (https://terra.nasa.gov/) that describes the orbital degradation and attempts to extent Terra as long as possible.**

L33-34: Could you had some contextual information as to why glaciers in western North America have been excluded in these global studies? Also, on L33, the authors note recent studies that leverage laser altimetry and then cite Jakob and Gourmelen (2023). However, this study utilizes CryoSat-2 data, which is a radar altimetry, so this sentence should be revised.

**A fair point and now one that we addressed under Referee 1's comments. A good catch on the term 'laser altimetry'! We now omit 'laser' from this statement.**

L47-53: The reference Copernicus DEM is derived from TanDEM-X SAR data, but due to the penetration of the SAR signal into snow/firn/ice, it is unlikely that the surface elevations over glacier in this dataset represent the true surface glacier height – particularly in accumulation areas (probably a negligible problem in ablation areas). This is likely to be unavoidable, but can the authors provide some comment on this and how the penetration of the SAR signal is likely to impact the DEM generation. Are there any optically derived DEM sources that can be used?

**A fair point and one that we acknowledge in the paper here and in the discussion (lines 156-161) section of the paper. Based on the area-altitude distribution, we expect the total area impacted by penetration bias to be low (1-2 percent). The close correspondence between our estimates of COP30 - ICESat-2 mass change estimates with those derived from optical imagery (Hugonnet et al., 2021) also suggests that the potential bias of the penetration to be negligible. We did observe a significant seasonal bias in our analysis when we include all ICESat-2 data which we believe is due to the presence of seasonal snow.**

L02-16: Results – Can the authors comment at all on how the in situ mass balance data within the region compares with these results (mass balance records form Peyto, Place and Helm Glaciers)?

**This is an interesting question but one where the sparse number of laser shots for these glaciers does not allow us to directly compare our results from either approach (i.e. COP30/ICESat-2 or ICESat-2 /GEDI) since the altitude distribution of elevation change for a given glacier is low. The regional average mass change for the Southern Coast Mountains from both approaches is comparable (about 1.0 m w.e. loss per year) to mass change from these three glaciers, however.**

Page 4, L6: (Fig. 1 is missing a closing bracket.

**Corrected.**

P5: L24-27 – Here is the mention of the penetration bias in the SAR derived DEM. I suggest that this be moved into the description of that dataset (as identified above) so that I comes earlier in the text.

**This point is now brought up in the data description and discussion section of the paper.**

L25: There is some inconsistency here, earlier in the text it is TanDEM-X while here it is Tandem-X, check for consistency throughout.

**A good catch. We corrected this and now use 'TanDEM-X' throughout the paper.**

L17-29: Discussion and Conclusion: Here I would suggest that the authors be a bit more detailed in their descriptions about why this work is important. Glaciers in Western North America are often overlooked in the global assessments, but are key sources of water for communities and for agriculture in these regions. So, these dedicated and more detailed assessments that investigate these glacier changes in more precise detail within these regions are fundamentally important. I would suggest that the authors make this more apparent in the text.

**A valid point; we now add information that acknowledges this point both in the introduction and at the end of the discussion of our paper.**